# Effects of Solid Die Types in Complex and Large-Scale Aluminum Profile Extrusion

**Tat-Tai Truong [1,2], Quang-Cherng Hsu [2,*] and Van-Canh Tong [3]**

[1]  Department of Mechanical Engineering, Hung Yen University of Technology and Education, Dan Tien, Khoai Chau, Hung Yen 49000, Vietnam; cadcamtai@gmail.com

[2]  Department of Mechanical Engineering, National Kaohsiung University of Science and Technology, 415 Chien-Kung Road, Kaohsiung City 80778, Taiwan

[3]  Department of Ultra-Precision Machines and Systems, Korea Institute of Machinery and Materials, Yuseong-gu, Daejeon 34103, Korea; canhtong@kimm.re.kr

*  Correspondence: hsuqc@nkust.edu.tw

**Abstract:** Increasing customer requirements for quality and productivity in extruding aluminum products has led to the development of different types of extrusion dies. In this study, three different types of dies, including traditional flat die, pocket die and spread die were designed to extrude complex and large-scale solid profiles. The design parameters for these dies were used from actual extrusion experience. The results obtained from steady-state simulation such as velocity, temperature, extrusion force, and die deformation were used to assess the advantages and disadvantages of the dies. Transient simulations were performed to analyze the evolution of transverse weld in the pocket and spread dies. The effects of ram speeds on the related extrusion parameters were also investigated. The research results provide useful guides for designers and engineers in selecting these types of extrusion dies.

**Keywords:** solid die types; extrusion die design; pocket die; spread die; complex aluminum profile; large-scale aluminum profile; metal flow; die deflection; transverse weld; ram velocity

## 1. Introduction

Aluminum extrusion dies can be divided into three basic types: solid die, semi-hollow die, and hollow die, in which the solid and the semi-hollow dies are used for extruding solid profile products, and hollow die is used for fabricating hollow profile products [1,2]. In the aluminum extrusion with the solid die, a hot billet is supplied into the container of the extruder and then pushed by a ram. The material flows through the die to form the product.

Die design plays a decisive role in the quality of extruded products. Unreasonable design can lead to a variety of product defects such as wave, bending, tearing, etc. [3–5]. Figure 1 shows the typical product defects when the extrusion dies are not designed properly. So far, the design of extrusion dies is commonly based on the experiences of engineers and expensive die testing processes [6,7]. Hence, the die design still is a challenging task. Moreover, one of the most important factors of the modern extrusion process is the transverse weld formed between new billet material and old material in the cavity of the die. Therefore, to ensure the quality of extruded products, the extrusion die must be designed to provide the necessary pressure for the transverse weld forming orifice [8,9], as well as to maintain continuity of extrusion. Because the complexity of the extrusion process, numerical simulation is commonly used to predict the process performance, which will aid for the design of the extrusion dies. Presently, with the development of computer and simulation software, complex

parameters of extrusion process can be obtained such as extrusion velocity, product temperature, die deformation, die stress, welding seam etc. [10–12].

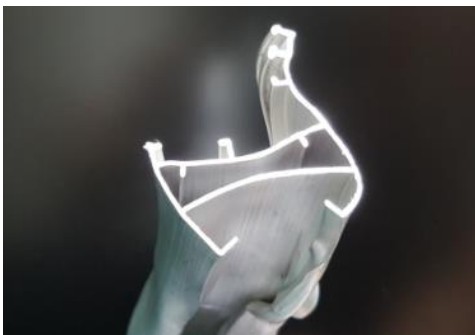

**Figure 1.** Product defects with an improper extrusion die design. The defects concern with significant bending and twisting of the profile.

There are many parameters involved in the die design. Among them, the parameters of bearing length were considered by many designers [2,13,14]. Using variable bearing length is one of the traditional methods for adjusting the balance of material flow [2,14]. The adjustment results usually rely on the experience of the engineer because of the complexity of the natural flow in the die. Several studies have introduced the methods for determining the appropriate bearing length. Miles et al. [13] proposed some standards and rules for the bearing length calculation based on their practical experiences. Lee et al. [15] used the numerical simulations to formulate an equation of bearing length in flat die hot extrusion. The bearing length equation is a function of the cross-sectional thickness and distance from the die center. The relationship between minimum bearing length and flow pressure was studied by Van Ouwerkerk [16]. Viswanath et al. [17] proposed a method for calculating bearing lengths of simple solid geometry using numerical simulations. This method was verified by extrusion experiments for traditional flat-face die. In general, the calculation of bearing length is relatively complicated and time-consuming, especially in the case of complex cross-sectional profiles [16,18].

Besides the bearing length, the design of solid extrusion dies also concerns with the geometries of the entrance and the pocket [14]. The pocket extrusion die provides the pressure needed for charge weld [2,14]. On the other hand, using the pocket in front of the die exit is an effective solution for controlling the distribution of the velocity of metal flow in the die [2,14,19,20]. The pocket geometry can be designed with single or multiple steps. Multiple cavity pocket is more advantage compared to single cavity one [21]. The design of the pocket has been considered by many authors. Li, Q et al. [22] investigated the effect of the angle and volume of the cavity on metal flow in the pocket extrusion die using 2D finite element (FE) analysis. They showed that adjusting the pocket angle is an effective solution to balance the flow. Li, Q et al. [23] revealed the effect of the entrance geometry and the pocket structure on flow in the solid die using simulation. They pointed out that using multiple steps can reduce the volume of the dead-metal zone, and the entrance geometry of the pocket can be used to eliminate the concave effect formed on the flat surface of a profile after extrusion. Fang et al. [21] studied the influence of pockets in the two-cavity die extrusion employing three-dimensional (3D) FE simulations. They showed that the designed die with multi-step pockets can reduce extrusion pressure and distribute the flow more evenly. Lee et al. [24] studied the optimal pockets' geometry of solid die for H-shaped aluminum profiles. The entrance geometry of the cavity was described by a B-spline curve, in which the control points of the curve were modified to achieve optimal geometry with the least deviation of velocity. Zhang et al. [25] designed the pocket geometry for complex aluminum profile production using an automatic mesh deformation method. Xavier [19] compared the relevant extrusion parameters of the flat-face, and the feeder dies by a 3D FE analysis. They indicated that the metal flow and temperature in extruded products are more evenly distributed when using the die with a feeder.

When the size of the extrusion product is larger than the container diameter, the spread dies can be used [14]. The spread dies are usually equipped with a spreader plate placed in front of the die orifice, which is responsible for adjusting the flow expansion to provide enough material to the die opening. Similar to the pocket die, the spread die creates the necessary pressure to ensure that the transverse weld is maintained during continuous extrusions. The design of the spread die for a large and complex profile is more difficult compared to the traditional flat and pocket dies. The main obstacles for the designer are the entrance geometry and the configuration parameters of the die so that the material can be distributed evenly, and to ensure the die strength. Kim et al. [26] optimised the opening geometry of the spread die for extruding a flat panel profile with a size larger than the container diameter of the extruder. Liu et al. [27] studied the types of inlets for the spread die of solid aluminum profile. They showed that the fan-shaped entrance is consistent with the flat plate profile, with this type of entrance, better extrusion parameters can be achieved in terms of temperature, velocity distribution in extrusion profiles, and die deformation. Imamura et al. [28] conducted an experimental extrusion of a simple flat profile. They reported that the extrusion force of the spread die was reduced by about 30% compared to the traditional flat die.

From the above literature review, the design parameters of dies and die types play an important role in ensuring product quality and extrusion productivity. The pocket and variable bearing length have been widely used in the die design for solid profiles. However, most studies have focused on simple aluminum profiles. There has been a lack of design guidelines for solid extrusion dies, and an in-depth comparison between the three types of solid extrusion dies for complex geometry profiles has not been reported.

In this paper, three types of dies for extruding complex, large-scale aluminum profiles were studied. To balance the metal flow in the extrusion product, the dies were designed with a 3D model using practical experience parameters. After that, numerical simulations of extrusion were conducted by HyperXtrude 2017 software (Altair Engineering, Inc., Michigan, USA) with the Arbitrary Lagrangian-Eulerian (ALE) algorithm. The steady-state extrusion parameters, such as extrusion force, velocity, temperature, and die deformation, were investigated. The transient simulation is utilized to explore the evolution of transverse weld in the billet-to-billet extrusion process of the pocket, and the spread dies. Finally, the effects of ram speeds on the related extrusion parameters were examined. The advantages and drawbacks of these die types are analyzed and evaluated based on the obtained parameters from simulations.

## 2. Die Design

In this study, the cross-section geometry of the product is asymmetric, the wall thickness varies from 2.0 to 5.0 mm, and the main body region of the product attaches four snap-fit elements. Figure 2 shows the 3D model (Figure 2a) and the essential dimensions of the geometry of the extruded product (Figure 2b). Stenger and Laue [29] evaluated the complexity of product geometry through a geometric classification of extrusion profiles, where solid profiles were classified into seven groups from the simplest geometry group (group A) to the most complex geometry group (group G). The product profile in this study is in group F, which has geometries with abrupt and thin wall cross-sections or wide cross-sections. Therefore, the 3D model shown in Figure 2 is a complex geometry profile. The maximum height and width of the products are 240.35 and 40 mm, respectively. Based on the largest size of the product profile, the traditional flat die and pocket die are selected to extrude on a machine with a container diameter of 310 mm (corresponding to a 4000-ton extruder); the spread die is chosen to extrude on an extruder with a container diameter of 186 mm, (corresponding to an 1800-ton machine).

Figure 3 shows the layout of the die exit of the flat-face die and the 3D model of the flat-face die. Die structure consists of the main die plate, die bearing region, and three run-out steps. The maximum diameter and height of the die are 360 and 90 mm, respectively. The maximum taper angle of run-out

is 3°. The geometry of these steps is determined empirically to ensure the die strength and to provide enough safety when the extruded metal flow does not touch the run-out surfaces.

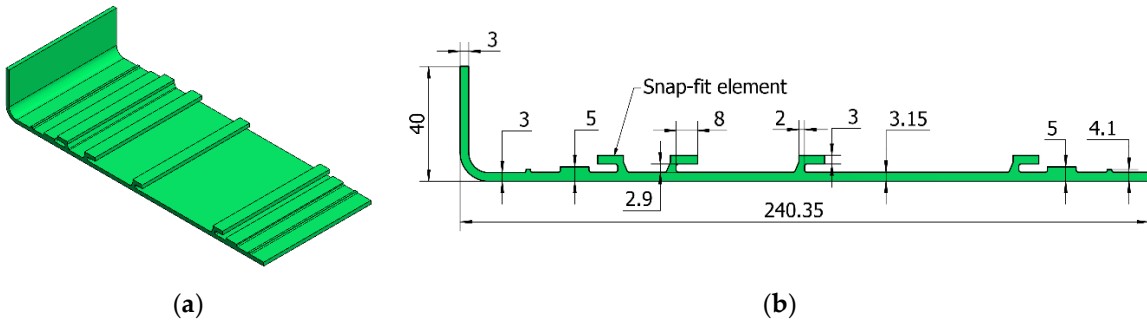

(**a**)    (**b**)

**Figure 2.** The geometry of the extruded product: (**a**) The 3D model; (**b**) The cross-section geometry with essential dimensions of the product (unit: mm).

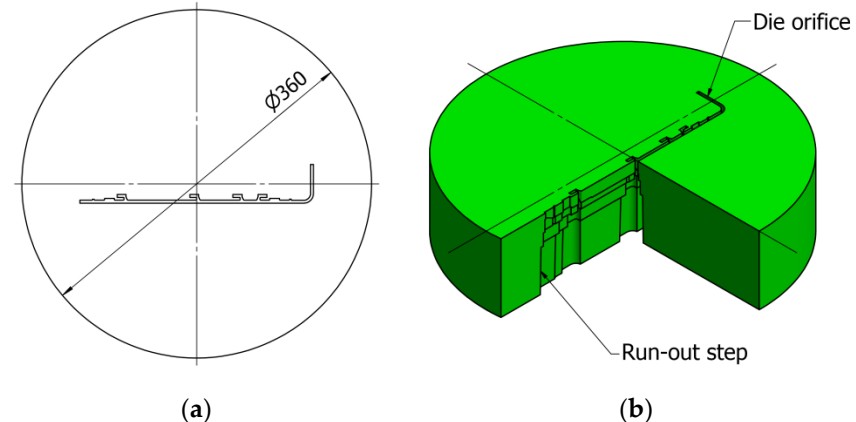

(**a**)    (**b**)

**Figure 3.** Design of the traditional flat die: (**a**) Layout of the die exit; (**b**) the 3D model of the flat-face die (unit: mm).

Figure 4 displays the layout for the die orifice with basic dimensions, and the 3D model of the pocket die. The maximum diameter and height of the pocket die are designed similarly to the traditional flat die, in which the depth of the cavity is 32 mm; the minimum distance of the entrance is 32 mm; the maximum circle diameter of the die entrance is 280 mm. The run-out of this die is also designed with the same geometry and structure as the traditional flat die.

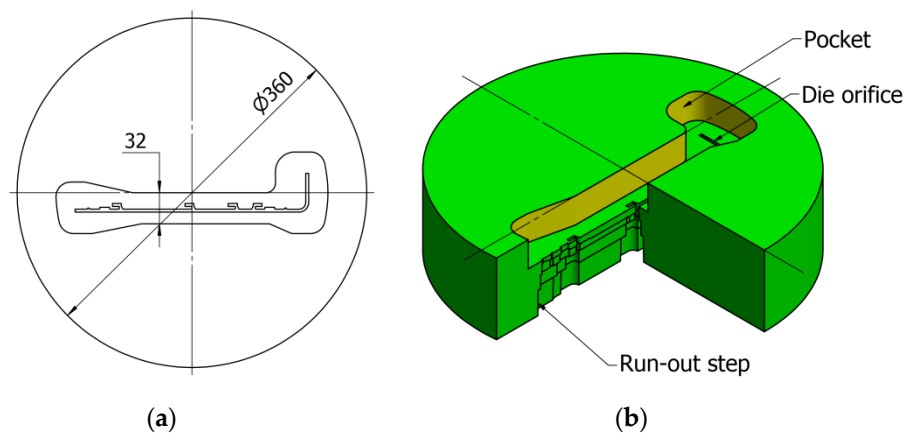

(**a**)    (**b**)

**Figure 4.** Design for the pocket die: (**a**) Layout for the entrance; (**b**) Simplified 3D model (unit: mm).

Figure 5a shows the structure, and main dimensions of the spread die. Figure 5b shows the 3D model of this die. With the same cross-section geometry of the product, the spread die uses the billet with a diameter smaller than the size of the product, which is the main difference in this die design compared to other dies. Therefore, it is necessary to use appropriate tactics to distribute the material evenly in the extrudate. Previous studies have shown that the structure and entrance geometry plays a vital role in the spread die design [26]. Thus, this led to the fact that this die design becomes very complicated. Although the design of spread die is complicated, this kind of dies offer some advantages over the other die types. In practice, designers and engineers often modify the entrance geometry of spread die design to meet the customer requirements. It is necessary to provide a suitable die structure and incorporate the use of supporting solutions such as bearing length adjustment or pocket to achieve a high-quality product. The spread die is designed with a maximum diameter of 360 mm, in which the maximum height of the leading die plate (lower die) and the spreader die (upper die) are 90 and 90 mm respectively, the maximum spread angle of the upper die is 30°. The upper die consists of a step with a depth of 30 mm placed before the die entrance. Moreover, a pocket is designed in front of the die orifice of the lower die to support flow correction. The maximum circumferential circle diameter of the material flow through the die is 280 mm. Design for run-out is similar to the two dies mentioned above.

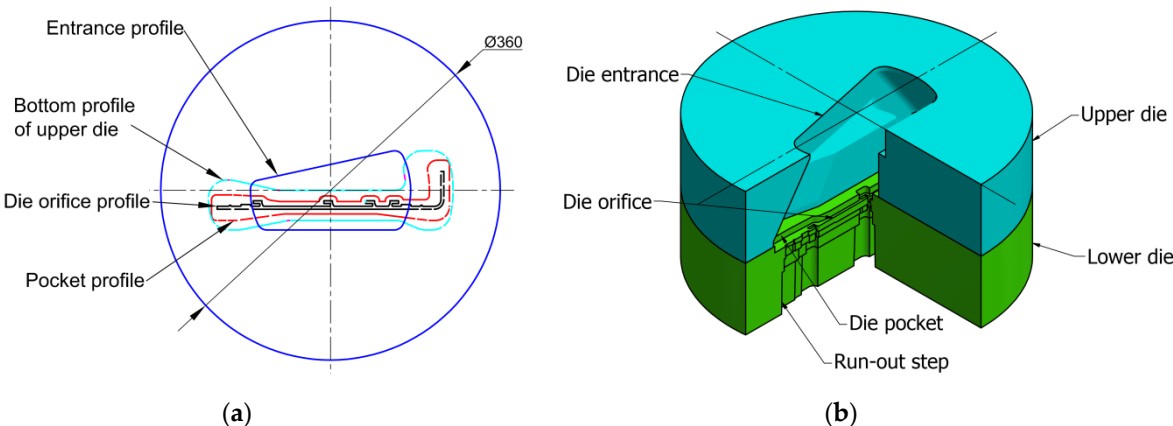

(**a**)　　　　　　　　　　　　　　　　　　　　　　　　　　　　　　(**b**)

**Figure 5.** Design for the spread die: (**a**) Layout of the die; (**b**) Simplified 3D model (unit: mm).

After the preliminary designs, the bearing length will be calculated. In this study, the calculation of the maximum bearing length ($B_{max}$) is based on an empirical formula, which relates to the width of the product as follows:

$$B_{max} = K_w t, \qquad (1)$$

where $K_w$ is the coefficient of the effect of the die orifice width (or product thickness) and $t$ is the main width of the die orifice (in this study $t = 3.15$ mm). The value of $K_w$ is from 2.0 to 3.0, which is often used in practice with a pocket combination. According to Miles et al. [13], $K_w$ from 1.4 to 3.5 were recommended. For the investigation purpose, this study adopts the values of $K_w = 2.5$ for calculating the bearing lengths of the initial extrusion dies.

Because of the complexity of the cross-section geometry of the product, variable bearing lengths are used to balance material flow. Some experience indicators are used in this study, as listed below:

1. The bearing area further away from the center will have a smaller length value.
2. Bearing lengths for regions with complex geometries (such as grooves for assembling, branches with changing directions) will be shorter than that for adjacent areas (multiplied by a coefficient from 0.75 to 0.9).
3. The bearing length at the tip of the product profile is calculated to be approximately 0.6 times smaller than the adjacent bearing.

Calculation scheme for bearing length is shown in Figure 6. Product profile is divided into 18 primary regions. The maximum bearing length value is in the area closer to the center (region 7 with the wall thickness of the product $t = 3.15$ mm). Calculated bearing values for the regions of the dies are listed in Table 1.

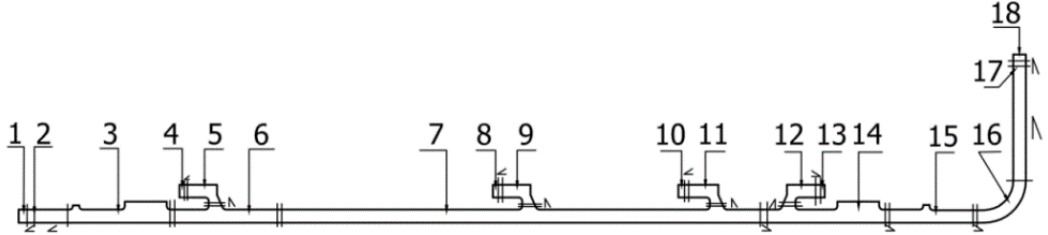

**Figure 6.** Calculation scheme for bearing regions of the dies.

**Table 1.** The bearing lengths in the regions for the extrusion dies.

| Bearing Lengths with $K_w$ | Bearing Region | | | | | | | | | | | | | | | | |
|---|---|---|---|---|---|---|---|---|---|---|---|---|---|---|---|---|---|
| | 1 | 2 | 3 | 4 | 5 | 6,14 | 7 | 8 | 9 | 10 | 11 | 12 | 13 | 15 | 16 | 17 | 18 |
| ($K_w = 2.5$) | 3.1 | 5.2 | 5.9 | 3 | 5.3 | 6.9 | 7.9 | 3.9 | 6.5 | 3.6 | 5.8 | 5.7 | 3.4 | 5.5 | 3.9 | 3.4 | 2 |

## 3. Finite Element Modelling

Steady-state and transient simulations are carried out. The models of the dies created by CATIA V5R20 software ((Dassault Systèmes, Vélizy-Villacoublay, France) are saved to STP file format and transferred to the simulation environment by HyperXtrude 2017 software. All exposed material areas are extracted for simulating the extrusion process. Figure 7a–c show the FE models with the flat-face, pocket, and spread dies, respectively. The meshing process is automatically generated by the software, in which the selection of the fine mesh size is based on the software recommendation. Then, rough mesh with a size of 13 mm is assigned to the billet area. The mesh size decreases gradually in the spread and pocket chambers. Fine mesh with a size of 0.4 mm are created in the bearing region, where the large plastic deformation occurs. The element type used for the bearing and profile regions is the triangular prism, while the tetrahedral element type is suitable for the other areas. The total numbers of elements used for the flat die, pocket die, and spread die are about 1,300,000, 900,000, 1,400,000, respectively.

In this study, the billet material is AA6063. The continuous equation with the Sellar–Tegart model is used to describe the behavior of flow stress of AA6063 shown in Equation (2):

$$\sigma = \frac{1}{\beta}\sinh^{-1}\left(\frac{Z}{A}\right)^{\frac{1}{n}}, \tag{2}$$

where $\sigma$ indicates the flow stress of the material; $\beta$ and $A$ are the material coefficients; $n$ is the exponent; $Z$ is the Zener–Hollomon coefficient and is calculated by formula Equation (3):

$$Z = \dot{\varepsilon}e^{Q/RT}, \tag{3}$$

where $\dot{\varepsilon}$ is the effective strain rate; $Q$, $R$, and $T$ are the activation energy, gas coefficient, and absolute temperature, respectively.

Material parameters of the constitutive equation are listed in Table 2. The material of the tools (ram and container) and the die is H13. The physical and thermal parameters of AA6063 and H13 are shown in Table 3. The extrusion parameters such as container diameter, billet length, billet temperature, die and tools temperature, and ram speed are shown in Table 4. The simulations were conducted on the HP-Z420 workstation. The average time for steady-state simulations of the traditional flat,

the pocket, the spread die were about 4, 1.4, and 2.5 hours, respectively. The average time for transient simulations for the pocket and the spread die were roughly 31.1 and 42.2 hours, respectively.

**Table 2.** Material parameters of the constitutive equation [30].

| β (m² MN⁻¹). | A (s⁻¹) | N | Q (J · mol⁻¹) | R (J mol⁻¹ · K⁻¹) |
|---|---|---|---|---|
| 0.04 | $5.90152 \times 10^9$ | 5.385 | $1.4155 \times 10^5$ | 8.314 |

**Table 3.** Physical and thermal parameters of the billet and tools material [4].

| Material | Density (Kg/m³) | Young Modulus (GPa) | Poisson's Ratio | Thermal Conductivity (W/m.K) | Heat Capacity (J/(kg °C)) |
|---|---|---|---|---|---|
| AA6063 | 2700 | 40 | 0.35 | 198 | 900 |
| SKD61 | 7870 | 210 | 0.35 | 24.3 | 460 |

**Table 4.** Extrusion parameters in simulation of different die types.

| Extrusion Parameters | Traditional Flat Die | Pocket Die | Spread Die |
|---|---|---|---|
| Container diameter (mm) | 310 | 310 | 186 |
| Billet length (mm) | 200 | 200 | 200 |
| Extrusion ratio | 72.5 | 72.5 | 26.1 |
| Billet temperature (°C) | 480 | 480 | 480 |
| Container temperature (°C) | 450 | 450 | 450 |
| Die temperature (°C) | 450 | 450 | 450 |
| Ram speed (mm/s) | 1, 3, 5, 7, 9 | 1, 3, 5, 7, 9 | 1, 3, 5, 7, 9 |

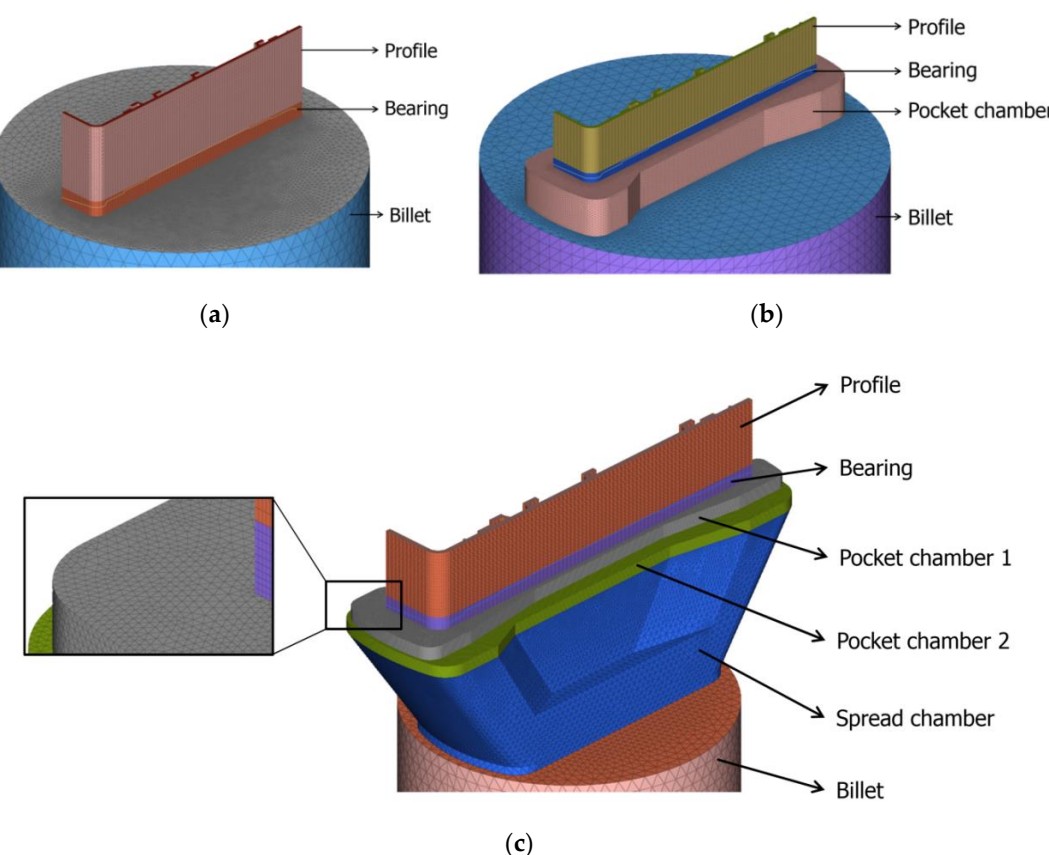

**Figure 7.** Meshing for the extrusion dies: (**a**) The flat-face die; (**b**) The pocket die; (**c**) The spread die.

## 4. Results and Discussion

### 4.1. Extrusion Velocity of the Product

The goal of the extrusion die design is to ensure the velocity in the extruded product is evenly distributed, and it is a decisive factor in the quality of the extruded product [31,32]. The velocity relative difference (VRD) shown in Equation (4) is used to assess the overall level of the velocity distribution of the extruded product [10,31,33].

$$VRD = \frac{\sum_{i=1}^{n} \frac{|V_i - V_a|}{V_a}}{n} \times 100\% \tag{4}$$

where $V_i$ is the speed at node $i$; $V_a$ is the average velocity of all nodes; $n$ is the number of nodes considered in cross-section of product. The smaller the VRD value, the better the flow distribution. Smaller VRD also means that product quality is better. The value of VRD for balancing metal flow is defined by the designer, which is expected when the die design is as small as possible. In this study, if VRD is less than 2% of the metal flow is considered to be balanced [11,27].

Figure 8 shows the flow distribution in the extrudate of the conventional flat die obtained from simulation with bearing length calculated by Equation (1) with $K_w = 2.5$. The velocity distribution on the product is uneven in this case. Velocity tends to be faster and uniform in the region near the center and slow in the region far from the center (region 1 and region 2). The velocity in region 1 is slowest, which can be explained by the obstruction formed by the complex geometry. The difference between the maximum and minimum velocity in the product profile is 20.37 mm/s. The calculated VRD value is 4.23%. However, this value is too large and needs to be further improved.

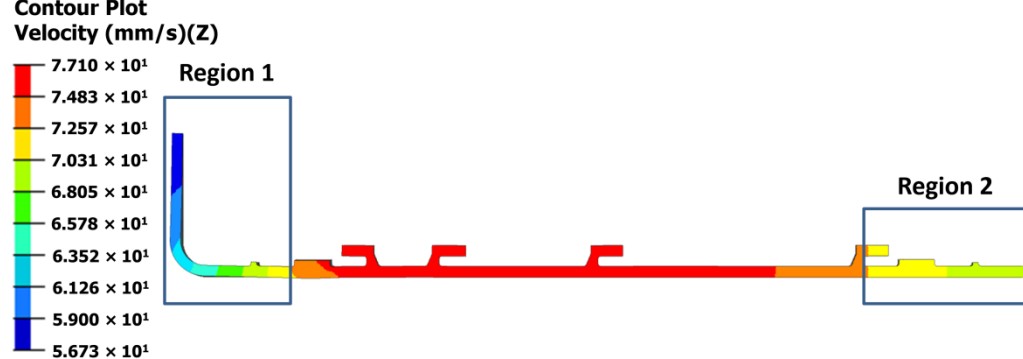

**Figure 8.** Velocity distribution in the extrudate of the traditional flat die with the calculated bearing length of $K_w = 2.5$.

To balance the metal flow in the extruded product of traditional flat die, the bearing length needs to be increased. Then $K_w$ coefficient should be increased accordingly. Table 5 lists bearing lengths of the regions in traditional flat dies with different $K_w$ values. The simulation results of velocity distribution for $K_w$ coefficient of 3.5 and 4.5 are shown in Figures 9a and 9b. It is observed that the differences between the maximum and minimum velocity are reduced from 10.37 to 5.37 mm/s. In other words, increasing $K_w$ leads to the fact that the material flow becomes more balanced. Hence, the traditional flat die must use a coefficient of $K_w = 4.5$ (corresponding to the maximum bearing length of 14.2 mm) to balance the metal flow in the extrudate.

Figure 10 indicates the velocity distribution in the pocket and spread dies. It shows that the velocity distribution with spread die is nearly even (Figure 10b). In this case, the calculated VRD value is 0.43%. The geometry of the product is not warped or bent. With a maximum bearing length of 7.9 mm ($K_w = 2.5$), the material flow can be balanced. However, controlling the metal flow in the spread die is more difficult compared to the other dies.

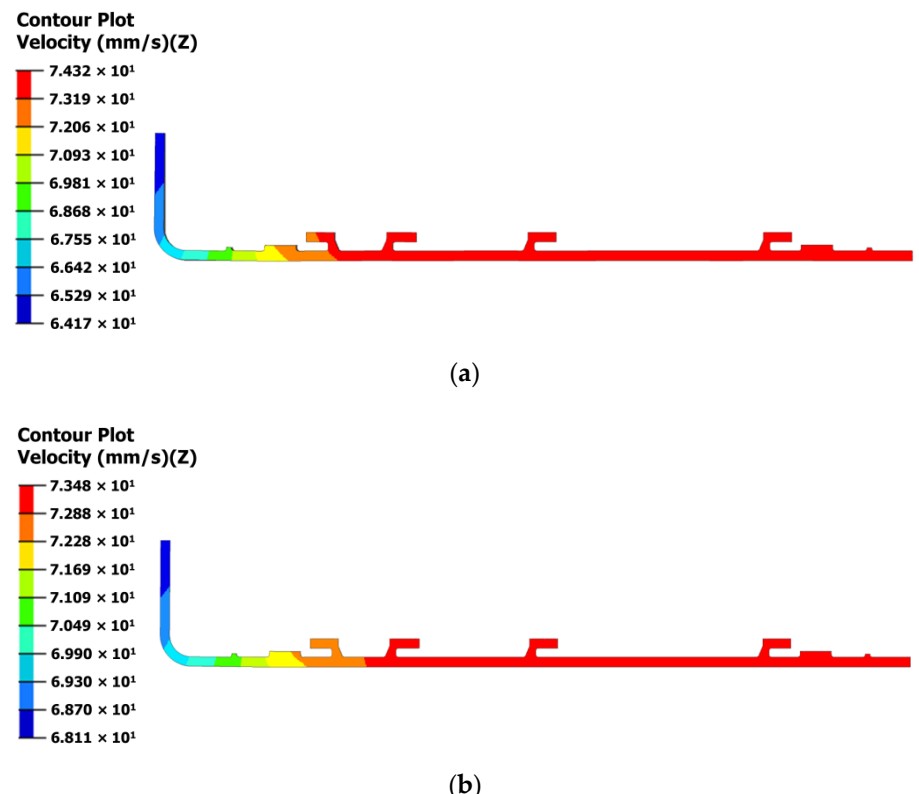

**Figure 9.** Velocity distribution in the products of the traditional flat dies with different $K_w$ coefficients at the ram speed of 1 mm/s: (**a**) $K_w$ = 3.5; (**b**) $K_w$ = 4.5.

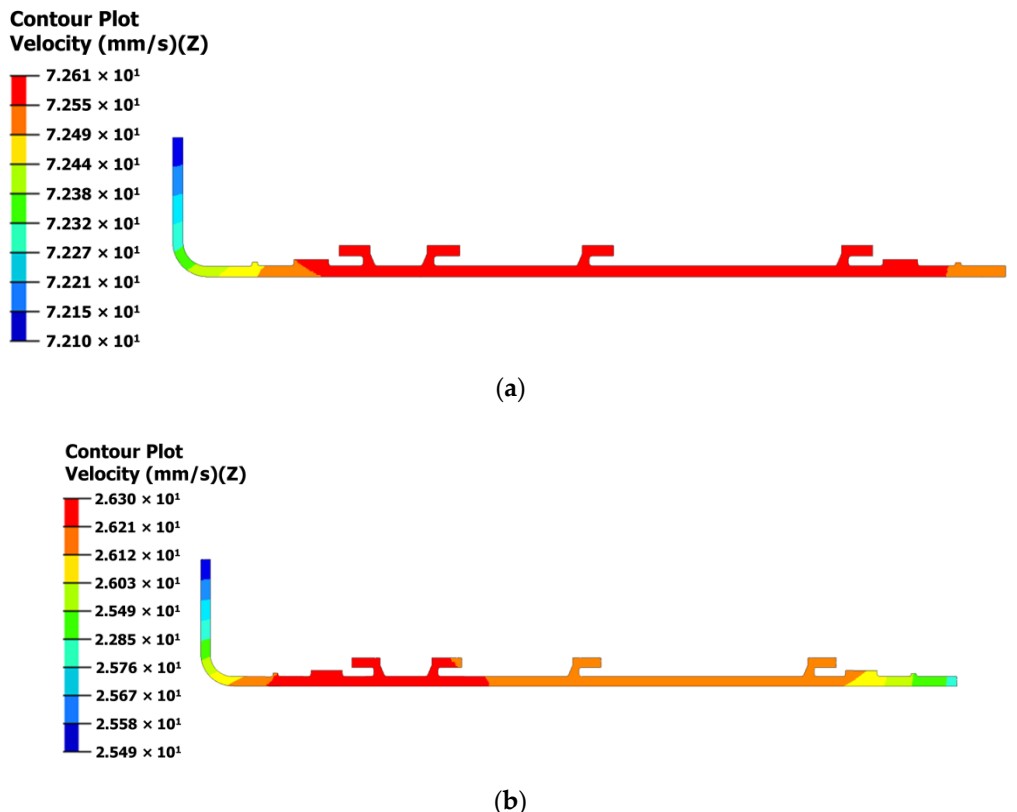

**Figure 10.** Velocity distribution in the products at the ram speed of 1 mm/s of the dies: (**a**) The pocket die; (**b**) The spread die.

**Table 5.** Bearing lengths for the regions with different $K_w$ coefficients.

| Bearing Lengths with $K_w$ | Bearing Region | | | | | | | | | | | | | | | | | |
|---|---|---|---|---|---|---|---|---|---|---|---|---|---|---|---|---|---|---|
| | **1** | **2** | **3** | **4** | **5** | **6,14** | **7** | **8** | **9** | **10** | **11** | **12** | **13** | **15** | **16** | **17** | **18** |
| ($K_w$ =3.5) | 5.2 | 8.6 | 9 | 4.4 | 7.3 | 10 | 11 | 5.5 | 9.1 | 4.8 | 8 | 7.9 | 4.7 | 8.5 | 6.2 | 5.4 | 3.2 |
| ($K_w$ =4.5) | 7.1 | 11.8 | 12.2 | 5.8 | 9.6 | 13.2 | 14.2 | 7 | 11.7 | 6.3 | 10.5 | 10.4 | 6.2 | 11.5 | 8.5 | 7.4 | 4 |

To further investigate the effect of ram speed on velocity distribution on the profiles of different die types, a series of steady-state simulations were conducted with ram speeds of 3, 5, 7, and 9 mm/s. Figure 11 demonstrates the changing trend of VRD with the ram speeds. In the case of the pocket die, the velocity remains almost balance with a slight vary in VRD when changing the speed of ram. With the traditional flat die, the velocity tends to be more uniform with the increasing ram speed, indicated by a significant decrease of VRD. For the spread die, the velocity distribution becomes more uniform when the ram speed is low (at 1 mm/s) or high speed (at 9 mm/s). At the ram speed of 3 mm/s, VRD of the spread die is maximized. The changing trend of VRD of the spread die when increasing the speed of ram up to 3 mm/s can be explained by the factors that cause the impact of the die structure and the effect of braking the metal flow by the dead-metal zone.

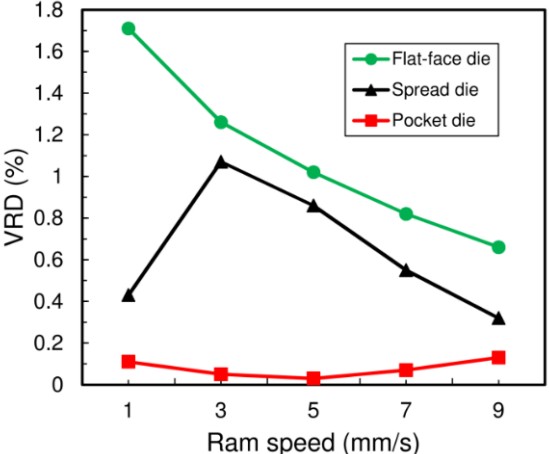

**Figure 11.** Influence of ram speeds to velocity distribution in the product of the extrusion dies.

*4.2. Extrusion Force*

Extrusion force is one of the crucial parameters in designing dies. It is also a criterion to evaluate the rationality of die design and is the basis for selecting the appropriate extruder [30]. Excessive extrusion force increases wear and can destroy the die, and this affects die life and cost [34].

Figure 12 shows the effect of ram speed on the required extrusion force. One can see an increasing linear trend of extrusion force. Hence, increasing ram speed negatively affects the extrusion process. With increasing the ram speed from 1 to 9 mm/s, the extrusion force is increased about 370 tons by using traditional flat and pocket die, and about 126 tons by using the spread die. This increased amount of extrusion force is indispensable because it will require extra work for the deformation process. At the same ram speed, the pocket die requires the highest extrusion force, which is about 2.5 times larger than that of the spread die. That can be explained by the increase in the extrusion ratio. Moreover, at the same speed, the extrusion force required by the pocket die is always higher than that of the traditional flat die. This is because using pocket die results in higher friction between the extruded material and the die walls.

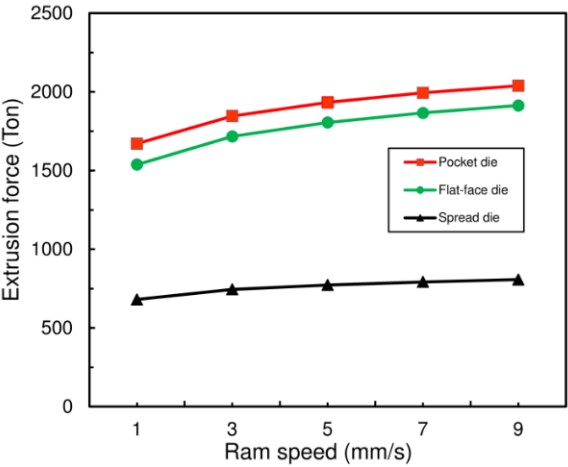

**Figure 12.** Effects of ram speed on the extrusion force in three die types.

### 4.3. Extrudate Temperature

The evolution of temperature in aluminum extrusion die is of great interest for both designers and manufacturers. Because heat can affect the mechanical properties and microstructure of the product, it is an essential criterion for evaluating the validity of the designed die. In practical extrusion, increasing extrusion speed is always desired by manufacturers because it increases productivity. However, this also leads to an increase in the temperature of the extruded product [35]. If the temperature exceeds the melting point of the material, surface defects may occur [34,36]. With AA6063, the maximum extrusion temperature of the desired product is less than about 560 °C [27]. In this study, the standard deviation of temperature (SDT) is used to evaluate the temperature distribution in the product:

$$SDT = \sqrt{\frac{\sum_{i=1}^{n}(T_i - T_a)^2}{n}}, \tag{5}$$

where $T_i$ is the temperature at node $i$; $T_a$ is the average temperature of all nodes; $n$ is the number of nodes considered in a cross-section of the product. Figure 13 shows the temperature distribution in the extruded products at a ram speed of 1 mm/s. It can be seen that the temperature distribution for the traditional flat die is more uniform than those of other dies. This is attributed by two reasons: 1) the contact area between flat die and material in die exit region is less than that of other dies and 2) the bearing length in this area of the traditional flat die is longer than the other dies leading to increased heat transfer. The temperature tends to reach its highest value in the middle part of the product, and minimal temperature occurs at the left or right edges of the profile. As a result, the region near the center experiences severe deformation. The maximum temperatures for the traditional flat, pocket, and spread dies are 497.5 °C, 504.9 °C, and 490.6 °C, respectively. The higher temperature results from the increased friction and work required for the deformation process corresponding to the die with more contact area and the larger accumulative volume of material flow.

Figure 14 shows the relationship between the maximum temperature and the ram speed for different extrusion dies. Figure 14 demonstrates that the temperature rises with the increase of extrusion speed. The temperature quickly increases in the initial speed range from 1 to 3 mm/s, but the increasing rate is diminished as the speed greatly increases. The total amount of increased temperatures in the flat, pocket, and spread dies are about 33 °C, 39 °C, and 59 °C, respectively. It is noted that at the ram speed of 9 mm/s, the maximum temperature of all three die types is less than 560 °C. Hence, the defects caused by the overheating problem can be avoided because maximum extrusion temperature is lower than the allowable extrusion temperature for AA6063 material.

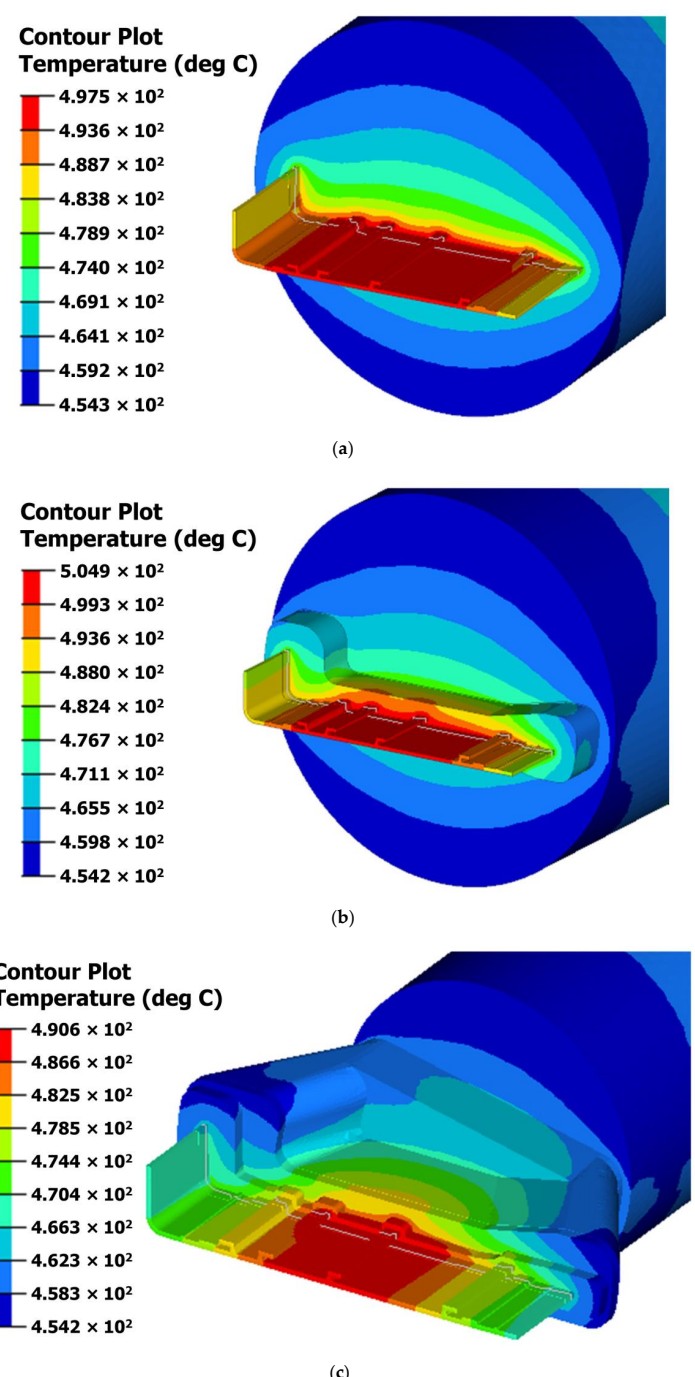

**Figure 13.** Temperature distribution in extrusion products of the dies, (**a**) The flat-face die; (**b**) The pocket die; (**c**) The spread die.

Figure 15 illustrates the effect of ram speed on the SDT and the difference between the maximum and minimum temperature (ΔT) in the extruded products of different dies. SDT and ΔT exhibit a general decrease tendency with increasing ram speed, except that the spread extrusion die shows a short increase in SDT and ΔT at low ram speed range. SDT and ΔT of the flat die are the smallest among the others. Moreover, SDT and ΔT of the spread die are minimal at the ram speed of 1 mm/s, while SDT and ΔT of the other two dies reach the minimum at the maximum ram speed of 9 mm/s. Therefore, increasing the ram speed has a positive effect on the temperature distribution in the products of the pocket and flat extrusion dies. This is because the temperature distribution becomes more uniform with increasing ram speed. However, increase ram speed also increases the maximum extrusion

temperature and required extrusion force as shown in Figures 12 and 14. Hence, the ram speed should be well controlled in practice.

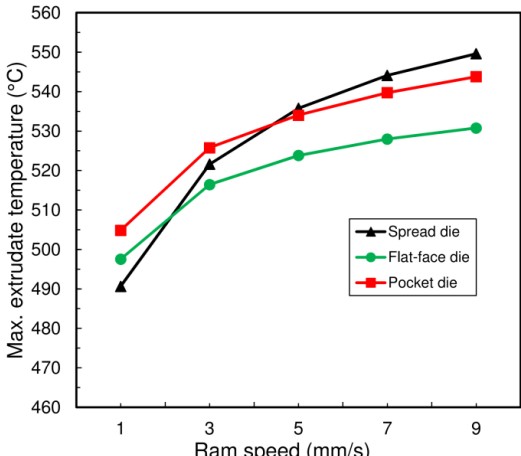

**Figure 14.** Effect of ram speed on the maximum extrudate temperature of the dies.

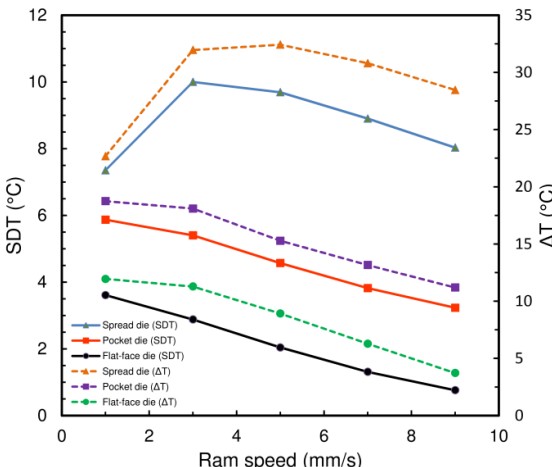

**Figure 15.** Effect of ram speed on temperature deviation in the product of solid extrusion dies.

### 4.4. Extrusion Die Deformation

The information of die deformation is valuable for die design because die deformation directly affects the geometry accuracy, tolerance of extruded product, and die life. Although the elastic deformation is inevitable, it is always expected that the deflections of the die are minimal. Based on the die deformation, the designer can adjust the design, such as modifying the cross-section of the product profile to minimize the deviations caused by the deformation.

Figure 16 denotes the deformation of the dies at a ram speed of 1 mm/s. It clearly shows that the die deformation is more intensive in the area closer to the center. The traditional flat die shows the largest deformation compared to the pocket die, and the spread die. The maximum deflections in the X, Y, and Z directions of the flat die are 0.08, 0.17, and 0.06 mm, respectively. Moreover, the maximum deformations in the directions of X, Y, Z of the pocket extrusion die are 0.06, 0.11, and 0.05 mm, respectively. The maximum deformations in the X, Y, and Z directions of the lower die of the spread die are 0.05, 0.1, and 0.09 mm, respectively. Thus, using additional pocket increases the extrusion force but reduces the pressure at the orifice region, which results in a reduction in the die deformation.

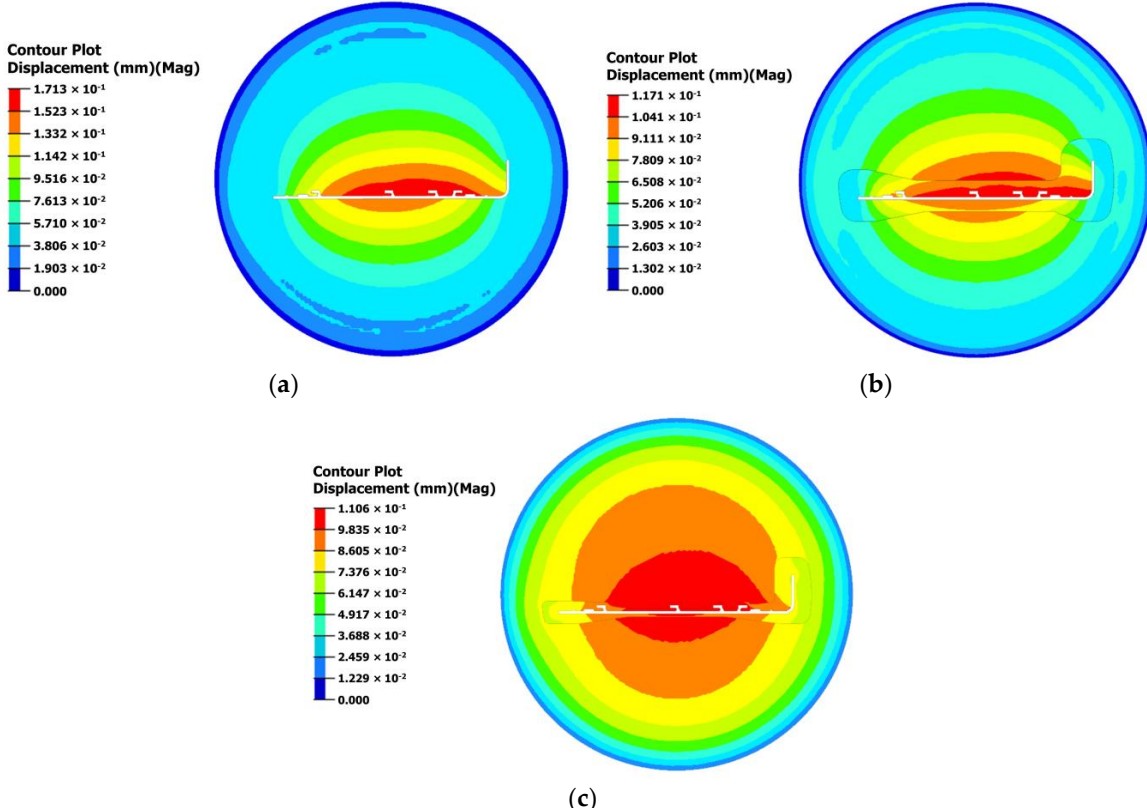

**Figure 16.** Die deformation: (**a**) Traditional flat die; (**b**) The pocket die; (**c**) The lower die of the spread die.

Table 6 summarizes the effect of ram speeds on the deformation of the dies. It is seen that increasing ram speed increases the deflection in the dies. The maximum deformation values of the flat die are highest compared to the other dies for all the speeds. On the other hand, the deformation of the dies tends to increase with increasing ram speed, except for the pocket die. It is observed that the structure of the pocket die can maintain deformation stability, and therefore, it is not affected by the increasing ram speed.

**Table 6.** Effect of ram speed on die deformation of solid extrusion dies.

| Max. Deflection of Dies (mm)(Mag) | Ram Speed (mm/s) | | |
|---|---|---|---|
| | 1 | 5 | 9 |
| Flat-face die | 0.17 | 0.18 | 0.19 |
| Pocket die | 0.12 | 0.12 | 0.12 |
| Spread die | 0.11 | 0.13 | 0.13 |

*4.5. The Transverse Weld of the Pocket Die and the Spread Die*

Pocket die and spread die are commonly used in billet-to-billet extrusion [9]. The pocket (or spreader) placed in front of the die orifice will control the flow and reduce the die deformation. It also acts as an element that controls the pressure needed for forming the transverse weld. Depending on the requirements of the customer, the part containing the charge weld may have to be removed with a high-quality product request.

In practice, the length of the transverse weld can be determined according to the position of the so-called "stop mark" [37,38] occurred after completion of the extrusion process of a billet. Figure 17 shows the stop mark, which is caused by the formation of friction between extruding material and bearing surfaces when stopping the ram.

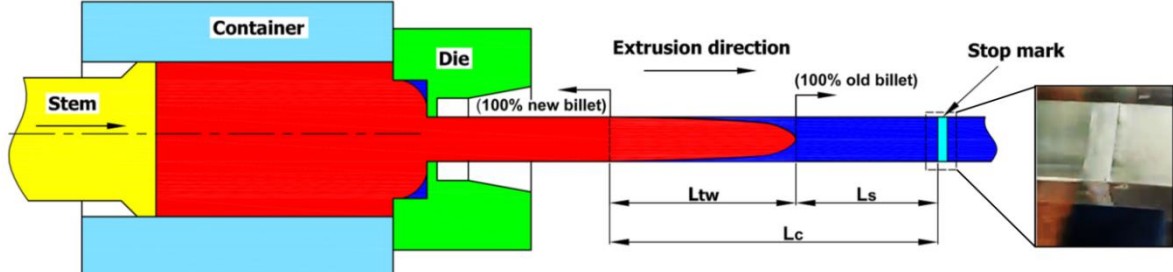

**Figure 17.** The appearance of stop mark and charge weld in the continuous extrusion of solid dies. The formation of a transverse weld and its length ($L_{tw}$) with distance from stop mark to starting position of charge weld ($L_s$), and distance from stop mark to the cut-off location at the end of charge weld ($L_c$).

Figure 18 shows the invasion (replacement) of the new material stream in the pocket and spread dies in different time steps. The initial interface of the transverse weld is flat (time step = 0 s)) and then gradually bent (see Figure 18a (time step = 8 s) and Figure 18b (time step = 30 s)). This figure also indicates that materials in the dead-metal zone are strongly influenced by sticky friction. The pocket extrusion process takes only 60 seconds, while the spread extrusion process takes 250 seconds to fulfil 100% new material.

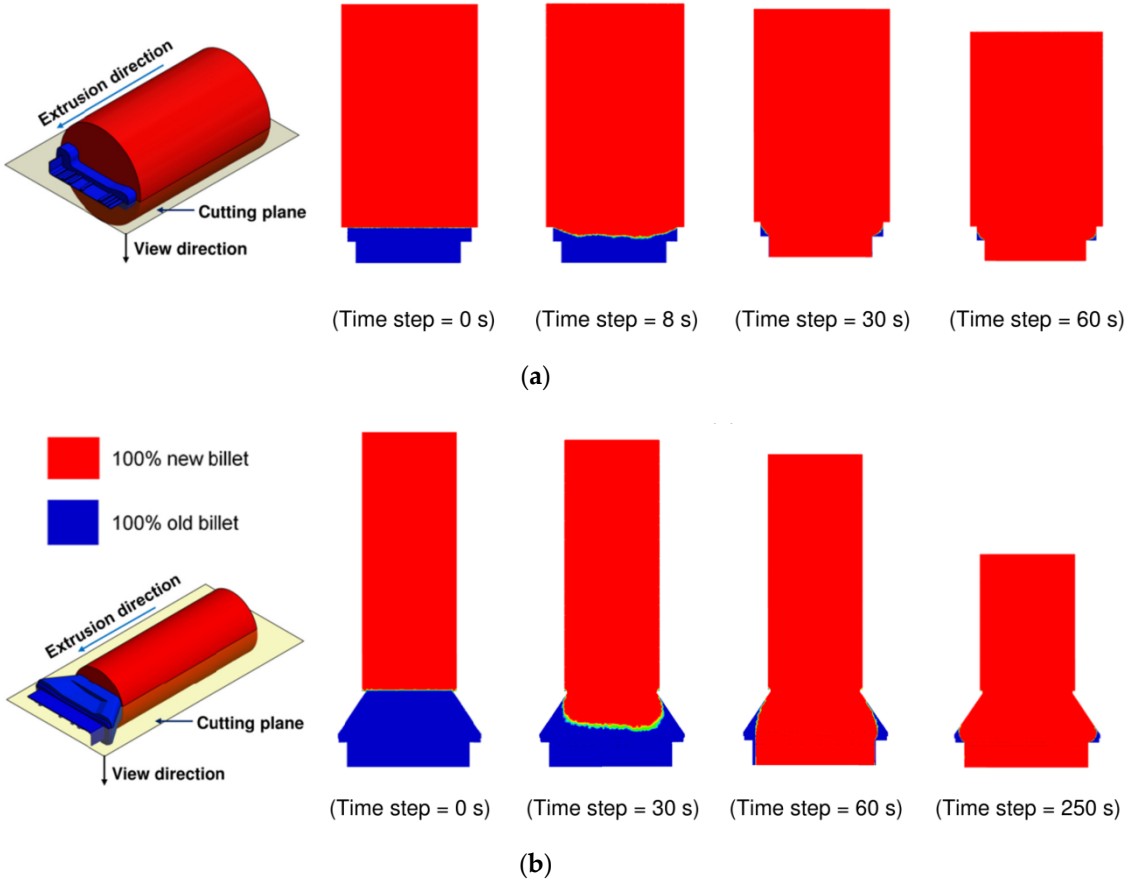

**Figure 18.** Progression of the charge weld in the dies: (**a**) The pocket die; (**b**) The spread die.

Figure 19 demonstrates the relationship between the length of the charge weld line and the percentage of new material simulated at the ram speed of 1 mm/s. It is worth noting that the average length of the charge weld for the spread die is twice that of the pocket die. This is because the material

volume in the pocket chamber is less than the spreading chamber, and the material in the dead zone is harder to wash off.

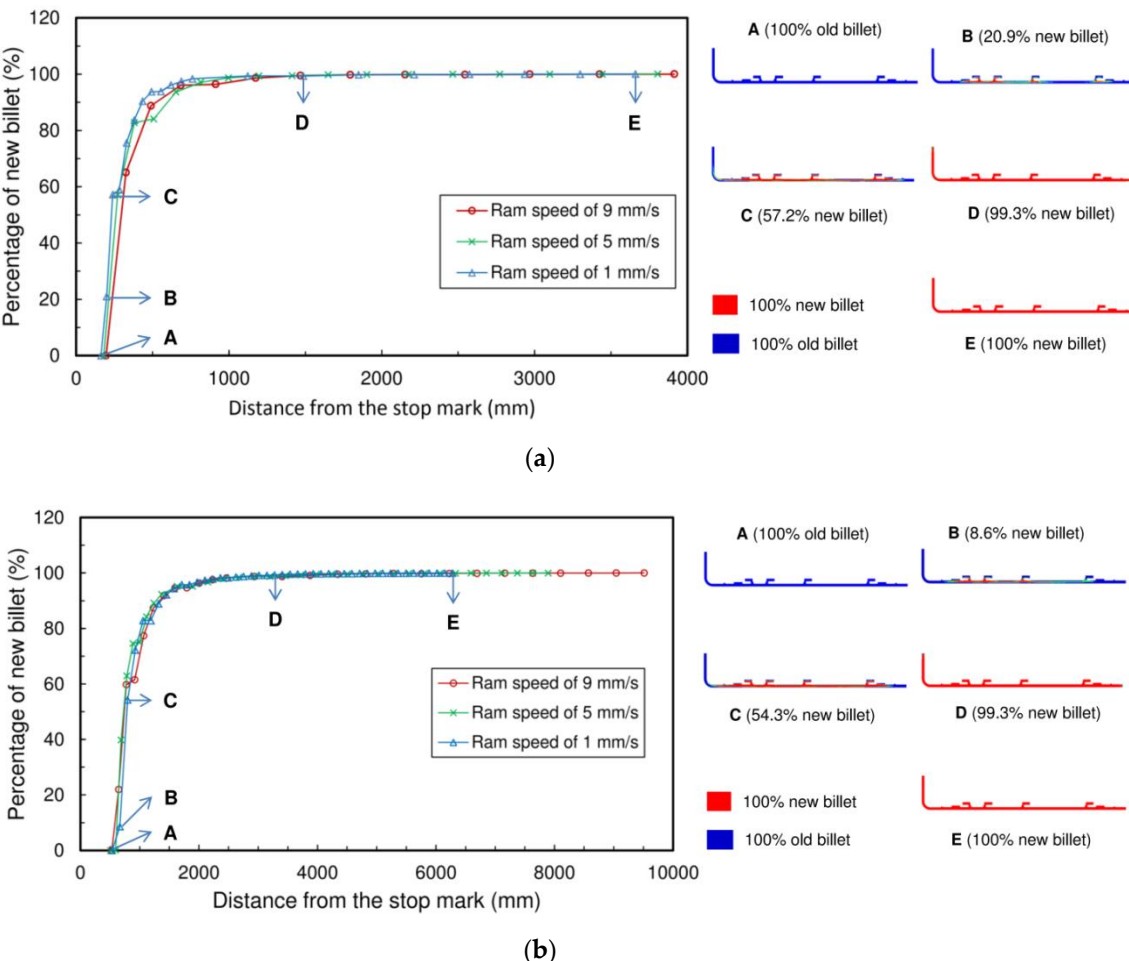

**Figure 19.** Percentage of the new material with the distance of the charge weld in the extruded product of the dies: (**a**) The pocket die; (**b**) The spread die.

Table 7 shows parameters related to the charge weld at different ram speeds of 1, 5, and 9 mm/s. Table 6 clearly shows that increasing extrusion speed leads to an increase in the length of the transverse weld in both dies. The charge weld lengths of the two dies are small at low extrusion speed (at ram speed of 1 mm/s). In particular, the increase rate in the weld length of the pocket die is much slower compared to that of the spread die. The weld length increases by about 221 mm for the pocket die, while it is 3243 mm for the spread die.

**Table 7.** Transverse weld parameters with increasing ram speeds.

| Ram Speed (mm/s) | Length of Transverse Weld | |
|---|---|---|
| | **The Pocket Die (mm)** | **The Spread Die (mm)** |
| 1 | 3497 | 5742 |
| 5 | 3624 | 7385 |
| 9 | 3718 | 8985 |

## 5. Conclusions

In this study, three different types of solid dies used complex profile extrusion were designed. Steady-state and transient simulations were conducted using HyperXtrude 2017 software based on the

ALE algorithm. A series of parameters obtained from simulation such as metal flow pattern, extrusion force, extrudate temperature, die deformation, and transverse weld length were analyzed. The effect of ram speeds on extrusion parameters was also evaluated. The conclusions of this study can be drawn as follows:

1.  Using the traditional flat die has the advantage of achieving a more uniform temperature distribution in the extruded product when compared to other solid dies. The extrusion force of this die is about 5% smaller than that of the pocket die. This type of die has a simple structure and is easy to fabricate. However, the needed bearing length is longer than other solid die types, which is undesirable in extrusion practice. Moreover, the deformation of the flat-face die is the largest, the maximum die deflection at a speed ram of 1 mm/s up to 0.17 mm; This may exceed the permissible tolerance of the product. Deformation reduction solutions for this die need to be considered further. When using the flat-face die, a low ram speed can be used to reduce die deformation. Finally, using the traditional flat die is not suitable for extruding complex profiles.

2.  Using the pocket extrusion die has many advantages compared to other types of solid dies, the velocity is more balanced; the deformation is minimal; transverse welding length is half shorter when compared to the spread die. Moreover, when the ram speed increases, the value of VRD changes little and the die deformation is maintained stably, and the welding line length does not increase significantly (about 221 mm when increasing the ram speed from 1 to 9 mm/s). This die type is the best choice to ensure quality and productivity in extruding aluminum products. However, the pocket and flat-face dies have the disadvantage of being unable to extrude products with sizes larger than the container diameter of the extruder.

3.  Using the spread die has the advantage that the required extrusion force is the smallest when compared to other solid dies, which is about 40% of the extrusion force required for pocket extruders with the same product profile. It is an extremely effective solution when it is necessary to reduce a large amount of extrusion force or extruding products on small machines with product sizes larger than the extruder diameter. This die type has been growing rapidly in recent years because it can increase the flexibility of production. However, this die type is difficult to design to be able to uniformly distribute the flow in extruded products. Hence, using a pocket and variable bearing lengths is necessary to balance the metal flow for extruding complex product profiles. The more complicated die structure makes the fabrication cost bigger than other types of solid dies. The difference in temperature distribution in the extrusion product of this die is much larger than in other cases, which can be overcome by the increase in temperature for the die plates. The spread extrusion forming the transverse horizontal welding length is twice as long as the pocket die, and the charge weld length is increased rapidly with increasing ram speed (about 3243 mm when increasing the ram speed from 1 to 9 mm/s).

**Author Contributions:** T.-T.T. came up with ideas, synthesized data, designed and simulated extrusion dies, and wrote the original manuscript; Q.-C.H. has instructed, supervised and edited the content; V.-C.T. analyzed the results, reviewed and edited the manuscript. All authors have read and agreed to the published version of the manuscript.

**Funding:** This research was partially funded by Ministry of Science and Technology, Taiwan, under grant number MOST 108-2221-E-992-066-.

**Acknowledgments:** The authors would like to acknowledge J-J Sheu, Department of Mold and Die, National Kaohsiung University of Science and Technology for providing the use of HyperXtrude 2017 software and valuable discussion.

**Conflicts of Interest:** The authors declare no conflict of interest.

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
