# Peer review of "Effects of Solid Die Types in Complex and Large-Scale Aluminum Profile Extrusion"

_applsci, doi:10.3390/app10010263_

Round 1

Reviewer 1 Report

This manuscript describes the effects of solid die types in complex and large-scale aluminum profile extrusion. The manuscript is well organized. My comments, concerns, and questions are as follows.

Why did the authors select the 3D model shown in Fig. 2 as one of complex geometry profiles? Please add reasons in the die design section. It should be helpful for readers to understand the following results and discussion section.

Please add a front view of the 3D model in Fig. 2 to recognize the all scales easily.

Line 148: before the of the die entrance ==> before the die entrance

Please use the italic font for variables such as t, T, Q, and so on.

Line 189: Zeller ==> Zener

Line 277: speed ==> temperature

Line 277: velocity ==> temperature

Author Response

Reviewer 1:

This manuscript describes the effects of solid die types in complex and large-scale aluminum profile extrusion. The manuscript is well organized.

Thank you very much for your comments.

Why did the authors select the 3D model shown in Fig. 2 as one of complex geometry profiles? Please add reasons in the die design section. It should be helpful for readers to understand the following results and discussion section.

Answer: Thank you very much for your valuable comments. In this study, we aim at investigating the extrusion of complex and large-scale aluminium profile with three types of dies. The complexity of geometry profiles are commonly classified into seven groups from the simplest (group A) to the most complex group (group G). The product profile in this study is in group F, which has geometries with abrupt and thin wall cross-sections or wide cross-sections. Therefore, the 3D model shown in Figure 2 is a complex geometry profile. We add some more sentences to discuss this issue in the revised manuscript, are shown in line 115-123 on page 3.

Please add a front view of the 3D model in Fig. 2 to recognize the all scales easily.

Answer: As your suggestion, we add the front view of the 3D model into Fig. 2 in the revised manuscript, are shown in Figure 2b on page 4.

Line 148: before the of the die entrance ==> before the die entrance

Answer: We correct this mistake in the revised manuscript, are shown in line 156-157 on page 5.

Please use the italic font for variables such as t, T, Q, and so on.

Answer: We now use the italic font for all variables, are shown in lines: 198-201, 221-22, 291-292.

Line 189: Zeller ==> Zener

Answer: We correct this mistake in the revised manuscript, are shown in line 199 on page 6.

Line 277: speed ==> temperature

Answer: We correct this mistake in the revised manuscript, are shown in line 292 on page 11.

Line 277: velocity ==> temperature 

Answer: We correct this mistake in the revised manuscript, are shown in line 292 on page 11.

Reviewer 2 Report

It is a great paper, that give industrial solutions to a big problem in casting of metallic products.

Author Response

Reviewer 2:

It is a great paper, that give industrial solutions to a big problem in casting of metallic products.

Answer: We thank for your encouragement.

Reviewer 3 Report

Referee report for Applied Sciences

Title: Effects of solid die types in complex and large-scale aluminium profile extrusion

Authors: Tat-Tai Truong, Quang-Cherng Hsu, Van-Canh Tong

Summary

Present work studies the extrusion of an aluminium profile trough different solid die types with Finite Element Methods, presenting the results of the Finite Element analysis for each die type considered. The results are intended as a user guide for designers and engineers on this field of expertise

COMMENTS

The paper is interesting, well written and presented but the novelty of this paper is not clear.

Broad comments

Comment 1:  The motivation of the study is not clear. This work is presented as a useful guide for designers and engineers but no conclusions on the matter have been included. The conclusion section is more as a results exposition section. The authors don’t conclude on an ideal or optimal solution, just expose the results obtained in each variable analysed. The title is according to the work presented but the objective presented in the abstract is to ambitious comparing it with the conclusions presented. 

Specific comments

Comment 3: Line 45, “the parameters of bearing length were considered by many designers” Can the authors provide any reference? Some works are discussed in the same paragraph but this statement also needs to be provided with some references.

Comment 4: Line 60, “On the other hand…” That statement needs references too. Also, paragraph starting on line 81

Comment 5: Line 142, if the use of this kind of dies complicates the design, why is taken into account for this study? Aren`t the designers and engineers trying to avoid its use?

Comment 6: Line 155, “(or product thickness), t is the main…” Should be separated by “and” and no “,”

Comment 7: Line 157, authors select a 2.5 Kw coefficient, between 2.0 a 3.0. Why? This is not explained in the text. Also, table 1 present the coefficient 2.5, 3.5 and 4.5. Even though the authors selected previously kw=2.5, they work with different coefficients along the paper. Why? (line 226)

Comment 8: Line 179, no mesh study has been implemented? How the authors now that the mesh implemented is the optimal one?

Comment 9: Line 216, reference needed.

Comment 10: Line 249-250, why is that abrupt change justify?

Comment 11: Line 279, “it can be seen that the temperature distribution for the traditional flat die is more uniform than those of other dies”. Why? Explanation is required.

Comment 12: Line 291-293. The temperature reaches nearly 560C. That is not a problem even is close to the maximum tempertured?

Comment 13: Line 304, “therefore, increasing the ram…” Knowing that there are advantages and disadvantages when the ram speed is incremented, the authors do not express any compromise solution in that case. Any consideration here? The same in line 320-321.

Comment 14: Line 210-2011, “the early period of the shock wave” A time reference must be indicated since up to 0.08 ms Figure 7 shows more waves action on the sample boundary.

Comment 15: Line 325. The use of the expression “it seems” is unscientific in this case.

Comment 16: Line 330, the sentence is not corrected. It is unfinished.

Comment 17: The conclusion section must be improved to be a conclusion section. Now it only shows the results obtained but no decision whatsoever has been made over those results. The authors present the results but do not take solutions or measures according to those results. An analysis of the optimal solutions in each case would be lacking so that the article could fulfil the objective set out in the abstract, having interest for the scientific and industrial collective. Now the study feels incomplete and the novelty is not clear.

Author Response

Reviewer 3:

COMMENTS

The paper is interesting, well written and presented but the novelty of this paper is not clear.

Answer: Thank you very much for your valuable comments. We modify the manuscript to highlight the novelty aspect of our research.

Broad comments

Comment 1:  The motivation of the study is not clear. This work is presented as a useful guide for designers and engineers but no conclusions on the matter have been included. The conclusion section is more as a results exposition section. The authors don’t conclude on an ideal or optimal solution, just expose the results obtained in each variable analysed. The title is according to the work presented but the objective presented in the abstract is to ambitious comparing it with the conclusions presented. 

Answer: We modify the conclusion section to highlight the novelty aspect of our research. For more details, please see our answer for Comments 17.

“No comment 2”

Specific comments

Comment 3: Line 45, “the parameters of bearing length were considered by many designers” Can the authors provide any reference? Some works are discussed in the same paragraph but this statement also needs to be provided with some references.

Answer: As your suggestion, in the revised manuscript, we add a reference to support for the fact that the parameters of bearing length were considered by many designers, as shown in line 45-47 on page 2.

Comment 4: Line 60, “On the other hand…” That statement needs references too. Also, paragraph starting on line 81

Answer: As your suggestion, we also provide a reference to support for the correspondent statements in the revised manuscript, as shown in line 60 on page 2.   

Comment 5: Line 142, if the use of this kind of dies complicates the design, why is taken into account for this study? Aren`t the designers and engineers trying to avoid its use?

Answer: Although the design of spread die is complicated, this kind of dies offer some advantages over the other die types, for instance reducing extrusion force. Hence, it is still taken into consideration in our study. In practice, designers and engineers often modify the entrance geometry of spread die design to meet the customer requirements. In the revised manuscript, we add some more sentences to address this issue, as shown in line 150-152 on page 5.

Comment 6: Line 155, “(or product thickness), t is the main…” Should be separated by “and” and no “,”

Answer: In the revised manuscript, we replace “,” by “and” to avoid ambiguity, as shown in line 164 on page 5.  

Comment 7: Line 157, authors select a 2.5 Kw coefficient, between 2.0 a 3.0. Why? This is not explained in the text. Also, table 1 present the coefficient 2.5, 3.5 and 4.5. Even though the authors selected previously kw=2.5, they work with different coefficients along the paper. Why? (line 226)

Answer: Kw is often chosen based on the experience of engineer depending on specific extrusion case. For the investigation purpose, this study adopts some values of Kw of 2.5 for calculating the bearing lengths of the initial extrusion dies.

We rearranged Table 1 (Line 182) to avoid misunderstandings for the reader. With Kw = 2.5 is used for all die design initially, however, to emphasise one thing to balance the velocity with VRD less than 2%, Kw for the flat-face die needs to use Kw = 4.5. Therefore Table 2 shows the bearing lengths in the traditional flat die, as shown in line 242.

Comment 8: Line 179, no mesh study has been implemented? How the authors now that the mesh implemented is the optimal one?

Answer: The meshing process is automatically generated by the software, in which the selection of the fine mesh size is based on the software recommendation. In the revised manuscript, we now address this issue to avoid confusing, as shown in line 188-189.

Comment 9: Line 216, reference needed.

Answer: As your suggestion, we provide a reference in the revised manuscript, as shown in line 225-226.

Comment 10: Line 249-250, why is that abrupt change justify?

Answer: Because we use several discrete ram speeds such as 1, 3, 5, 7 and 9 mm/s, the abrupt change is observed at 3 mm/s. Therefore, we study two more ram speeds such as 2.0 and 2.5 mm/s. Such abrupt change is diminished, as shown in below figure.

Fig. A1. Influence of ram speeds to velocity distribution

Comment 11: Line 279, “it can be seen that the temperature distribution for the traditional flat die is more uniform than those of other dies”. Why? Explanation is required.

Answer: In the manuscript, we add explanation for the uniform temperature distribution of the traditional flat die. In general, this is caused by two reasons: 1) the contact area between flat die and material in die exit region is less than that of other dies, 2) the bearing length in this area of the traditional flat die is longer than the other dies leading to increased heat transfer, as shown in line 295-298 on page 11.

Comment 12: Line 291-293. The temperature reaches nearly 560C. That is not a problem even is close to the maximum tempertured?

Answer: As we have mentioned in the manuscript, because maximum extrusion temperature is lower than the allowable extrusion temperature for AA6063 material, the related problems caused by the overheating can be avoided. We further emphasize this issue in the revised manuscript, as shown in line 310-311 on page 11.

Comment 13: Line 304, “therefore, increasing the ram…” Knowing that there are advantages and disadvantages when the ram speed is incremented, the authors do not express any compromise solution in that case. Any consideration here? The same in line 320-321.

Answer: As your comments, in the revised manuscript, we add more sentences to discuss the compromise of increasing ram speed. As we observed in Figure 15, the increasing ram speed is beneficial for reducing the temperature standard deviation (SDT), and maximum difference between the maximum and minimum temperature (∆T), which implies that the temperature distribution becomes more uniform with increasing ram speed. However, increase ram speed also increases the maximum extrusion temperature and required extrusion force as shown in Figures 12 and 14. Hence, the ram speed should be well controlled in practice, as shown in line 323-326 on page 13.

Comment 14: Line 210-2011, “the early period of the shock wave” A time reference must be indicated since up to 0.08 ms Figure 7 shows more waves action on the sample boundary.

Answer: We do not understand your question clearly. Did you mean that Lines 210-211 in our manuscript? Then we have check that carefully and found no serious mistake.

Comment 15: Line 325. The use of the expression “it seems” is unscientific in this case.

Answer: We modify the expression to be scientific. In the revised manuscript, the sentence is changed to: “It is observed that the structure of the pocket die can maintain deformation stability, and therefore, it is not affected by the increasing ram speed”, as shown in line 346-348 on page 14.

Comment 16: Line 330, the sentence is not corrected. It is unfinished.

Answer: We modify the sentence to be: “Pocket die and spread die are commonly used in billet-to-billet extrusion”, as shown in line 351 on page 14.

Comment 17: The conclusion section must be improved to be a conclusion section. Now it only shows the results obtained but no decision whatsoever has been made over those results. The authors present the results but do not take solutions or measures according to those results. An analysis of the optimal solutions in each case would be lacking so that the article could fulfil the objective set out in the abstract, having interest for the scientific and industrial collective. Now the study feels incomplete and the novelty is not clear.

Answer: As your suggestions, we modify the conclusion section. Following conclusions are made:

Using the traditional flat die has the advantage of achieving a more uniform temperature distribution in the extruded product when compared to other solid dies. The extrusion force of this die is about 5% smaller than that of the pocket die. This type of die has a simple structure and is easy to fabricate. However, the needed bearing length is longer than other solid die types, which is undesirable in extrusion practice. Moreover, the deformation of the flat-face die is the largest, the maximum die deflection at a speed ram of 1 mm/s up to 0.17 mm; This may exceed the permissible tolerance of the product. Deformation reduction solutions for this die need to be considered further. When using the flat-face die, a low ram speed can be used to reduce die deformation. Finally, using the traditional flat die is not suitable for extruding complex profiles. Using the pocket extrusion die has many advantages compared to other types of solid dies, the velocity is more balanced; the deformation is minimal; transverse welding length is half shorter when compared to the spread die. Moreover, when the ram speed increases, the value of VRD changes little and the die deformation is maintained stably, and the welding line length does not increase significantly (about 221 mm when increasing the ram speed from 1 to 9 mms). This die type is the best choice to ensure quality and productivity in extruding aluminium products. However, the pocket and flat-face dies have the disadvantage of being unable to extrude products with sizes larger than the container diameter of the extruder. Using the spread die has the advantage that the required extrusion force is the smallest when compared to other solid dies, which is about 40% of the extrusion force required for pocket extruders with the same product profile. It is an extremely effective solution when it is necessary to reduce a large amount of extrusion force or extruding products on small machines with product sizes larger than the extruder diameter. This die type has been growing rapidly in recent years because it can increase the flexibility of production. However, this die type is difficult to design to be able to uniformly distribute the flow in extruded products. Hence, using a pocket and variable bearing lengths is necessary to balance the metal flow for extruding complex product profiles. The more complicated die structure makes the fabrication cost bigger than other types of solid dies. The difference in temperature distribution in the extrusion product of this die is much larger than in other cases, which can be overcome by the increase in temperature for the die plates. The spread extrusion forming the transverse horizontal welding length is twice as long as the pocket die, and the charge weld length is increased rapidly with increasing ram speed (about 3243 mm when increasing the ram speed from 1 to 9 mm/s).

Round 2

Reviewer 3 Report

The authors have made the required changes